# SGAN-IDS: Self-Attention-Based Generative Adversarial Network against Intrusion Detection Systems

**DOI:** 10.3390/s23187796

**Published:** 2023-09-11

**Authors:** Sahar Aldhaheri, Abeer Alhuzali

**Affiliations:** Faculty of Computing and Information Technology, King Abdulaziz University, Jeddah 21589, Saudi Arabia

**Keywords:** adversarial attacks, black-box attacks, generative adversarial networks, intrusion detection, offensive security

## Abstract

In cybersecurity, a network intrusion detection system (NIDS) is a critical component in networks. It monitors network traffic and flags suspicious activities. To effectively detect malicious traffic, several detection techniques, including machine learning-based NIDSs (ML-NIDSs), have been proposed and implemented. However, in much of the existing ML-NIDS research, the experimental settings do not accurately reflect real-world scenarios where new attacks are constantly emerging. Thus, the robustness of intrusion detection systems against zero-day and adversarial attacks is a crucial area that requires further investigation. In this paper, we introduce and develop a framework named SGAN-IDS. This framework constructs adversarial attack flows designed to evade detection by five BlackBox ML-based IDSs. SGAN-IDS employs generative adversarial networks and self-attention mechanisms to generate synthetic adversarial attack flows that are resilient to detection. Our evaluation results demonstrate that SGAN-IDS has successfully constructed adversarial flows for various attack types, reducing the detection rate of all five IDSs by an average of 15.93%. These findings underscore the robustness and broad applicability of the proposed model.

## 1. Introduction

An intrusion detection system (IDS) is a pivotal cybersecurity tool designed for both host and network monitoring. Its primary function is to discern malicious traffic from legitimate ones. In this realm, the network intrusion detection system (NIDS)—a subtype of IDS—continuously scans the network for malevolent activities and triggers alerts upon detecting suspicious traffic [1]. Such alerts typically convey details pertaining to the source address of the intrusion, the target or victim’s address, and the conjectured type of attack. Historically, network intrusion detection systems largely relied on signature-based methodologies. However, over the past two decades, the landscape has shifted toward anomaly detection strategies rooted in machine learning. Bolstered by recent advancements in artificial intelligence (AI) techniques, machine learning (ML) algorithms have been increasingly integrated into IDSs. Their incorporation, especially in recent years, has manifested as a potent and precise protective mechanism, registering commendable performance metrics. Several machine learning classifiers, encompassing decision trees [2], convolutional neural networks (CNNs) [3], support vector machines (SVMs) [4], artificial immune systems (AISs) [5], event-triggered H*∞* filters for nonlinear fuzzy systems [6], and Bayesian networks (BNs) [7], have found widespread adoption in IDS frameworks. Diverging from their traditional non-AI counterparts, these classifiers are adept at identifying suspicious traffic, unearthing novel patterns, and pinpointing anomalies in data.

Using a database of known attacks, testing systems to generate “benchmark” behaviors, and flagging any abnormality as a potential attack are common approaches used to train an anomaly-based IDS [8]. This kind of IDS is essentially an evaluator that uses past data to determine if network packets entering the system are malicious or not. In order to train an ML-based IDS properly, large amounts of data are required for that purpose. Data collection is typically a complex process because most of the data are private and subject to strict privacy policies [9]. However, when adding an IDS to a network, this has significant drawbacks. First, establishing a benchmark behavior in a dynamic environment like an IoT system may be difficult since devices are continually shifting, new devices are joining, and behaviors are changing. Second, protocols differ from one network to another. Third, as new cyberattacks increase, these systems become more vulnerable to unknown types of attacks. As a result, the detection process might be time-consuming since it requires data collection tailored to each system and attack type [10]. Therefore, it is vital to have a robust IDS that can detect zero-day attacks.

Improving the robustness and precision of an IDS entails minimizing (or ideally, eliminating) both false positives (erroneous alerts) and false negatives (overlooked intrusions), particularly concerning zero-day attacks. While the majority of IDS frameworks harness genuine network traffic to devise a detection model for identifying analogous threats, this approach alone might not suffice to augment the detection model’s accuracy. Furthermore, recent studies have highlighted a potential decrement in the detection and accuracy rates of IDSs when confronted with adversarial synthetic network flows [11,12]. Specifically, these synthetic datasets are derived from authentic malicious data.

Creating synthetic data is a significant problem that has interested researchers for several years. Previous research has used regression [13], classification [14], and Bayesian networks to sample from a combined multivariate probability distribution created by treating each column of a table differently [15,16]. Recently, generative adversarial networks (GANs) have gained popularity due to their ability to implicitly learn data distributions with arbitrarily complex dimensions. They have been investigated for the generation of synthetic data [17]. Researchers have recently produced several adversarial techniques, mostly in the domain of image classification. These methods are based on the notion of making minor modifications to the original input data in order for a machine-learning model to misclassify the data. This has proved to be extremely effective at creating artificial audio, images, and videos that seem realistic [18,19,20].

From another perspective, AI is a powerful tool that can be utilized by cybersecurity professionals to create cybersecurity solutions, and by attackers, who can take advantage of vulnerabilities in these IDSs to disrupt their detection mechanisms. Generative adversarial networks can be utilized to create nearly undetected adversarial attacks. Generating synthetic intrusion flows to simulate future attacks can be used to enhance the accuracy of IDSs and, therefore, improve the security of the whole network.

In order to improve IDS resilience, we propose a new generative adversarial network (GAN) framework, named SGAN-IDS, which generates malicious feature records for adversarial attacks on intrusion detection systems. The goal of the framework is to deceive the defense systems and carry out successful evasions in real-world networks. The framework, which builds upon GANs and self-attention mechanisms, includes a generator that creates adversarial malicious traffic records and a discriminator that learns from the BlackBox intrusion detection system (IDS) by analyzing its real-time outputs and providing feedback for the generator’s training. The IDS outputs can be obtained by querying it with traffic records. Experimental results have shown that SGAN-IDS successfully generated adversarial traffic flows, resulting in decreasing the detection rate, precision, recall, and F1 score of five BlackBox ML-based IDS models. The main contributions of this work are outlined as follows:We propose a novel framework called SGAN-IDS that uses adversarial training to make ML-based IDSs more resilient to attack detection.We introduce the idea of a self-attention mechanism to GANs to build adversarial traffic flows that will evade IDS detection.We evaluate the model using the CICIDS2017 dataset, which achieves highly accurate results.

### Motivation

In an era marked by escalating cyber threats, traditional intrusion detection systems (IDSs) are facing increasing challenges, especially from synthetic data vulnerabilities. However, the emergence of deep learning, particularly through generative adversarial networks (GANs), offers a promising countermeasure. Our framework taps into this potential, advocating for a proactive approach that harnesses AI’s strengths while addressing its vulnerabilities, setting a new standard for modern cybersecurity. Evolving threat landscapes: With cyber threats becoming more sophisticated, traditional IDSs often fall short. Our framework addresses the need for advanced solutions tailored to contemporary challenges.

Synthetic data vulnerabilities: Current IDSs can be bypassed by adversarial synthetic network flows. Our proposal directly confronts this vulnerability, aiming to enhance detection capabilities.Harnessing deep learning: The capabilities of GANs in understanding complex data distributions present an opportunity to revolutionize cybersecurity. Our framework leverages this potential for improved intrusion detection.Proactive approach: Our strategy emphasizes not just reacting to threats but proactively simulating potential attacks, ensuring IDSs are better prepared for real-world threats.AI in cybersecurity: As AI becomes integral to security solutions, it is crucial to address its potential vulnerabilities. Our framework seeks to turn AI’s challenges into strengths, enhancing overall security.

This paper is organized as follows. Section 2 provides a necessary background on the topic. Section 3 discusses the related work. While Section 4 provides an architectural overview of SGAN-IDS, and Section 5 contains details about the implementation. Section 6 describes the evaluation of our framework. Finally, we provide the conclusion in Section 7.

## 2. Background

### 2.1. Generative Adversarial Networks

In 2014, Goodfellow et al. [21] introduced the generative adversarial network (GAN) model, which is a type of neural network architecture used for generative modeling. A GAN model consists of two separate sub-models: a generator *G* and its counterpart, the discriminator *D*. These sub-models engage in unsupervised learning to learn the training data distribution and patterns in a way that the model could produce new data while keeping the training data features. The generator network must fight against an adversary in generative adversarial networks, which are based on a game-theoretic scenario. Samples are produced directly via the generator network. The discriminator network, on the other hand, attempts to tell the difference between samples from the training data (real data) and those from the generator (fake or generated data) (Figure 1).
(1)$$\begin{array}{cc} \min_{G}\max_{D}V\left(D,G\right)= & E_{X∼P_{\mathrm{dara}\;}\left(x\right)}\left[\log D\left(X\right)\right]\\ \; & +E_{z∼p_{z}\left(z\right)}\left[\log\left(1-D\left(G\left(z\right)\right)\right)\right] \end{array}$$

### 2.2. Attention Mechanism

The attention mechanism is inspired by the human brain’s attention process. When people look at photos, they do not examine every pixel. Instead, they selectively focus their attention on certain crucial portions of the image while dismissing the rest. In deep learning (DL), the attention mechanism serves to quantitatively determine the degree of focus among multiple system components. This mechanism was originally proposed in the field of image research in the mid-1990s. However, the major focus on attention mechanisms can be attributed to the Google DeepMind team, who incorporated attention mechanisms into recurrent neural networks (RNNs) for image recognition research [22].

The attention mechanism was first introduced by Bahdanau et al. [23] in the natural language processing (NLP) field. The purpose of their work is to overcome the bottleneck problem that might occur due to the presence of long sequences in automated language-to-language translations.

Attention is a component of network architecture that affects how input influences output. A query and a tuple of key–value pairs are transformed into output by an attention function. Weights are assigned to each value and are computed using a helper function with the relevant key input. The final result is a weighted sum of the input values [24].

#### 2.2.1. General Attention

In a neural network layer, this form of attention arises among various input elements. The general attention mechanism has three sub-aspects: the queries *Q*, the keys *K*, and the values *V* [23]. Given a query *q* and a set of key–value pairs (K,V), attention can be generalized to compute a weighted sum of the values based on the query and the related keys, as shown in the equation.
(2)$$A\left(q,K,V\right)=\sum_{i}\frac{\exp\left(e_{qk_{i}}\right)}{\sum_{j}\exp\left(e_{qk_{j}}\right)}v_{i}$$
where:*A* is the attention output.*q* is the query.*K* is the set of all keys.*V* is the set of all values.$e_{qk_{i}}$ represents the dot product between the query *q* and a specific key $k_{i}$. This dot product measures the compatibility of the query with that key.The term $\exp(e_{qk_{i}})$ is the exponential of the dot product, which amplifies the compatibility score.The denominator $\sum_{j}\exp\left(e_{qk_{j}}\right)$ is a normalization term. It sums the exponential scores for all keys, ensuring that the weights sum up to 1. This makes the mechanism probabilistic, as it essentially computes a weighted average of values.$v_{i}$ is the value associated with the key $k_{i}$.

Several researchers have investigated various general attention mechanism forms. Luong et al. [25] utilized “dot product attention”, where the value of attention was determined by a dot product of the weight and value. Another study used “scaled dot product attention”, where the attention values were scaled down in addition to a scaling factor [24]. Bahdanau et al. [23] investigated “additive” attention using the tanh function on the dot product.

#### 2.2.2. Self-Attention

Self-attention, also known as intra-attention, allows an encoder to attend to other parts of the input during processing (see Figure 2). Given the input matrix (*Q*, *K*, *V*), the self-attention matrix is constructed mathematically as follows:(3)$$\;\mathrm{Attention}\;\left(Q,K,V\right)=\mathrm{softmax}\left(\frac{QK^{T}}{\sqrt{d_{k}}}\right)V$$
where:*Q* represents the matrix of queries.*K* represents the matrix of keys.*V* represents the matrix of values.
Figure 2Self-Attention operation is presented graphically.
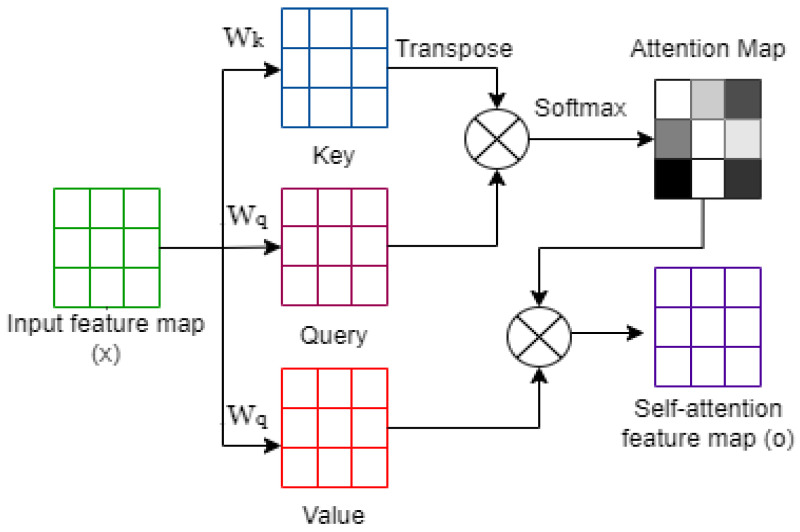



The concept of self-attention was initially introduced by Cheng et al. [26] in 2016. This work uses a modified long short-term memory (LSTM) network unit to implement self-attention. The LSTMN replaces the memory cell with a memory network, allowing for the storage of contextual representations of each input token in a single memory slot that grows in size over time until an upper-bound memory span is achieved. In 2017, Vaswani et al. [24] proposed a novel type of self-attention for both the encoder and decoder, allowing the transformer model to process all input words at once and represent the relationships between all words in a phrase. An improvement in the previous work led to the invention of bidirectional encoder representations from transformers by Devlin et al. in 2019 [27].

Moreover, in the fields of textual entailment [28] and video processing [29], the concept of self-attention has been effectively utilized.

### 2.3. Self-Attention for GANs

Self-attention generative adversarial networks are convolutional neural networks that utilize the self-attention paradigm to better synthesize new images by capturing long-range spatial correlations in existing images. Zhang et al. [30] proposed an approach where a self-attention mechanism was added to the GAN model to capture long-range, multi-level correlations in an image.

## 3. Related Work

Many research works have utilized deep learning techniques to produce synthetic data and enhance the accuracy performances of security-related detection tools. This paper focuses on the use of self-attention mechanisms and GANs to create synthetic adversarial attack traffic. Hence, we will shed light on some closely related research works.

Constructing adversarial data has been explored in areas such as malware detection [31,32,33]. For example, MalGAN [31] is a system that generates adversarial malware examples by using the GAN algorithm to bypass ML-based black-box malware detection tools. The developed model transforms malware samples into adversarial examples. A later work by Kawai et al. [32] sought to improve the work in [31] by integrating certain cleanware features (e.g., APIs) into the original malware. They used different learning techniques, different feature quantities, and a singular malware rather than many malware samples in the previous MalGAN, all to optimize performance. Similar studies have been conducted on other related security areas.

The following research studies focus on utilizing GAN to create adversarial network traffic, aiming to evade detection by intrusion detection systems.

Lin et al. [34] proposed a system denoted as IDSGAN; it aims to attack intrusion detection systems by constructing adversarial traffic records from malicious ones. IDSGAN utilizes the Wasserstein GAN algorithm [35], where the generator is used to produce malicious traffic records, and the discriminator layer is leveraged to classify and learn the malicious records from the benign ones by including a BlackBox IDS. The discriminator then provides the feedback to the generator for training purposes. IDSGAN was tested on the NSL-KDD dataset [36] and presented good results when combined with classification algorithms, such as SVM, naïve Bayes, MLP, and KNN. However, the authors of [12] indicated that IDSGAN has updated two functional characteristics of the tested network traffic records, contradicting the adversarial traffic generation requirements. Usama et al. [12] proposed another GAN-based approach that evades ML-based IDSs by generating adversarial traffic flows. The model was tested on the NSL-KDD dataset and can recognize only one type of malicious traffic attack, i.e., the probe attack. Our work, on the other hand, utilizes the self-attention mechanism to achieve better results in terms of accuracy and can construct many types of adversarial attack flows. Another line of work with similar goals is by Hydra [37]. The developed GAN-based model adds perturbations to specific network features, such as the packet rate and payload size. Their model addresses features related to DoS attacks. Therefore, their model cannot construct other types of adversarial attacks, unlike our work.

Charlier et al. [38] developed SynGAN, which is a framework that constructs adversarial attacks using the gradient penalty-Wasserstein GAN (GP-WGAN) algorithm. SynGAN generates a mutated synthetic distributed denial of service (DDoS) attack flow using real attack flows from NSL-KDD and CICIDS2017 [39] datasets. This work used the same dataset that we examined. However, their proposed framework focused on constructing specific adversarial flows while SGAN-IDS utilized self-attention in GANs to generate different types of attack flows. In a more recent work, Duy et al. [40] used Wasserstein GAN (WGAN) to generate adversarial attack patterns to evade ML-based IDSs in SDN. Their approach maintained the operational features of adversarial attacks by not altering the functional features of the original attack flows. Their proposed framework was tested on the NSL-KDD and CICIDS2018 [39] datasets and presented promising results. SGAN-IDS has the same goal, but its approach to achieving that goal includes the self-attention mechanism and it has excellent results.

Several research studies, such as ref. [30], have used self-attention mechanisms in GANs to produce synthetic data in computing domains, to capture correlations in images. To the best of our knowledge, self-attention mechanisms in GANs have not been utilized before to construct adversarial traffic flows that can elude detection by IDSs.

## 4. Method

In this section, we introduce the framework of a self-attention-based generative adversarial network against intrusion detection systems (SGAN-IDSs). Figure 3 depicts the overall architecture of our system, which consists of four main components: the generator *G*, discriminator *D*, attention layer, and BlackBox IDS. This framework has been trained to generate adversarial attack data from a GAN-based attack pattern.

In order to construct attack samples, the model uses GAN to create fake samples. These modified samples are added to the training dataset in order to build adversarial resilient models.

### 4.1. Problem Description

Intrusion detection systems are integral security components at the host and network levels. They detect malicious events such as malicious traffic and unauthorized access in hosts as well as networks. Enhancing the accuracy and robustness of an IDS can reduce (or eliminate) false positives (i.e., false alerts) and false negatives (i.e., undetected intrusions), especially for zero-day attacks. Most IDS models use real network traffic to create a detection model, which detects potential similar attacks. This may not be enough to improve the accuracy of the detection model of IDSs.

Generating synthetic intrusion flows to predict future attacks can be used to enhance the robustness of IDSs and, therefore, improve the security of the whole network. These adversarial data are constructed using GANs and self-attention mechanisms.

### 4.2. Generative Adversarial Networks (GANs)

In the following, we will describe how we utilize GANs in our system.

#### 4.2.1. Design of the Generator

The generator is responsible for generating adversarial data that are capable of misleading the IDS. It uses 10 latent features out of 79 features to generate traffic samples. The generator network has 4 layers: (1) an input layer, which has 10 neurons, (2) a first hidden layer, which has 32 neurons, (3) a second hidden layer, which contains 64 neurons, and (4) an output layer that consists of 79 neurons. Batch normalization, ReLu activation, and 20% dropout are applied to the input layer and hidden layers. For the output layer, SGAN-IDS utilizes linear activation (see Figure 4).

The discriminator will give the generator a score, and the weights will be adjusted to optimize the generator’s data generation. The generator loss is then determined using the discriminator’s classification; if it successfully manipulates the discriminator, it is rewarded; otherwise, it is penalized.

To train the generator, the following equation should be minimized:(4)$$L_{G}=E_{B\in R_{\mathrm{attack},Noise}}D\left(G\left(B,Noise\right)\right)$$
where $R_{\mathrm{attack}\;}$ denotes the original malicious traffic records, *G* denotes the generator, and *D* represents the discriminator. LG must be minimized in order to train and optimize the generator for deceiving the BlackBox IDS.

#### 4.2.2. Design of the Discriminator

The discriminator classifies both real and false data from the generator while it is being trained. The discriminator architecture has four layers: (1) an input layer, which is a size of 79 neurons, (2) a first hidden layer, which has 64 neurons, (3) a second layer that contains 32 neurons, and (4) the last layer, which consists of 1 neuron with sigmoid activation for the classification of the real vs. fake traffic sample. In our GAN model, we used the Adam optimizer with a 0.0001 learning rate for both the generator and discriminator (see Figure 5).

The discriminator attempts to maximize the following equation:(5)$$L_{D}=E_{r\in S_{\mathrm{normal}\;}}D\left(r\right)-E_{r\in S_{\mathrm{attack}\;}}D\left(r\right)$$
where *s* is the discriminator’s training set; $S_{\mathrm{normal}}$ and $S_{\mathrm{attack}}$ denote normal and adversarial traffic records, respectively, using predicted labels from the BlackBox IDS as ground truth.

### 4.3. Attention Model

This involves paying attention to a particular part of the input by weighing that part more than the other parts. During training, the model learns which parts of the input it should pay more attention to.

The backbone of the attention model is the multi-head attention layer, which takes into account the query, keys, and value-embedding to calculate the attention weights and context features of the input that can be used as input in the next layer for network traffic generation. Another main part of multi-head attention layers is the number of heads because it splits the embedding into the number of heads, calculates different attention outputs, and concatenates them, passing through a fully connected layer to produce the final attention context features and attention weights.

Multi-head attention layers also use scaled dot-product attention layers to calculate the context features and attention weights. Scaled dot-product attention layers also depend on the query, key, and value-embedding because they start with the multiplication of the query and key. Subsequently, softmax is applied after depth normalization to maintain the weights within a computational range. Following the multiplication of these results with the value, both attention weights and context features are obtained as output

The attention model approach is used to obtain accurate results in real-time inference; it increases the accuracy of the model in less time because it uses the same approach as humans, i.e., the human does not obtain knowledge of the whole picture in one go, the human starts to pay attention to a certain part and starts building their intuition, going through all the parts, obtaining knowledge. Attention models use the same approach by paying attention to a piece of the whole input and obtaining the idea after parsing all parts (see Figure 6).

### 4.4. BlackBox IDS

ML-based IDS demonstrates excellent performance in detecting attacks using non-adversarial data in the real world. Hence, in our system, we include an ML-based IDS in a black-box fashion. The BlackBox IDS will allow us to evaluate our constructed adversarial data flows in an unbiased manner as there will be no interference with the internal detection mechanism of the IDS.

We evaluated SGAN-IDS against five different machine learning-based IDSs: support vector machine (SVM), naïve Bayes (NB), multilayer perceptron (MLP), logistic regression (LR), and K-nearest neighbor (KNN).

## 5. Implementation of SGAN-IDS

The generator takes a noise vector as input and converts it into a fake training sample, which is subsequently passed via the discriminator. The discriminator uses both real samples (from the training data) and fake samples (produced by the generator) to try to distinguish between the two. The generator’s task is to produce malicious traffic that can mislead the IDS. We preserve the functional features of an original attack to confirm that the generative algorithm’s output is actually an attack. The dimensions of hidden layers are set based on the best outcomes. The architecture of the model is detailed in Section 4.

The training input contains normal traffic records and malicious traffic records. The malicious records are fed into the generator and transformed into adversarial records after adding noise. The output of the first hidden layer generator will be forwarded to the attention layer, which acts as *X* number of units, and two heads will be defined in the model. The adversarial malicious records and normal records are detected by the BlackBox IDS. It uses 79 features that have been preprocessed. Then it has the task of accurately predicting whether these data are benign or malicious traffic. For class identification, we adopt a threshold of 0.5 in this study. If σ≥0.5, IDS returns a value of 1 (attack); otherwise, if σ<0.5, the label is 0 (benign). After that, the discriminator uses the predicted labels to learn the BlackBox IDS. The generator and discriminator losses are estimated using the discriminator’s outputs and the IDS-predicted labels.

Algorithm 1 describes the procedure for training the SGAN-IDS.
**Algorithm 1** SGAN-IDS    **Input:** Normal and malicious features $\left(f_{1},f_{2},...,f_{N}\right)$;   **Output:** *G* outputs the network traffic     Init the hyperparameters of the generator (G)     Init the hyperparameters of the discriminator (D)     Init the state of the generator and discriminator     Init the state of attention (A)     Init the cost function (Q)     Input random array of normal distribution into *G*     Input *G*’s output and real output into *D*.     **repeat**          **repeat**             Discriminator Process          **until** *D* selects optimal hyperparameters          **repeat**             Fix the discriminator hyperparameter             Update *G* and *D* parameters             Update *A* parameters             Obtain results from the discriminator             Update cost function (Q)          **until** *G* selects optimal hyperparameters     **until** Epoch ends

Scikit-learn was used to implement all conventional machine learning algorithms. PyTorch is used as the deep learning framework in the experiment to implement the GAN model [41], which is trained for 10,000 epochs with a batch size of 30. We used the Adam optimizer with a 0.0001 learning rate for both the generator and discriminator.

## 6. Results

In this section, We introduce the dataset, evaluation criteria, and experimental setup, followed by the presentation of experimental results and comparative analysis.

### 6.1. Dataset

#### 6.1.1. CICIDS2017 Dataset

In 2017, the Canadian Institute for Cybersecurity released the CICIDS2017 dataset, which is an intrusion detection and prevention dataset. It includes updated real-world attacks as well as normal traffic [39]. The network traffic is analyzed by CICFlowMeter from Monday to Friday using information based on timestamps, sourced, destination IP addresses, protocols, and attacks [42]. It contains network traffic data collected from a simulated cyberattack scenario. The dataset is intended for use in the research and development of intrusion detection systems (IDSs) and includes both normal and malicious traffic. The data include a wide range of network protocols, including TCP, UDP, and HTTP, as well as different types of attacks, such as denial of service (DoS), probing, and malware. In the dataset, normal traffic is labeled as “benign” and attack traffic is labeled with the specific type of attack. The CIC-IDS2017 dataset can be used to train and evaluate IDS systems, as well as study the behaviors of different types of cyberattacks. Table 1 summarizes the statistics of attacks in the CICIDS2017 dataset.

#### 6.1.2. NSL-KDD Dataset

The NSL-KDD dataset is a popular dataset used for intrusion detection in computer networks. It is a refined version of the original KDD Cup 99 dataset, which aimed to address some of its limitations, such as the imbalance of classes and the presence of redundant data. The NSL-KDD dataset is designed to simulate a typical network environment, with a mix of normal and attack traffic, and it includes a wide range of network attacks, such as denial-of-service, unauthorized access, and probe attacks. The dataset is divided into a training set and a testing set, which allows for the evaluation of machine learning models for intrusion detection. The NSL-KDD dataset is widely used in research, and it has been used to evaluate various intrusion detection techniques, including artificial neural networks, decision trees, and rule-based systems. Additionally, the NSL-KDD dataset’s attack samples are split into four groups, with a total of around 125,000 records. This includes a mix of normal and attack connections; the specific number of records for each type of connection may vary depending on the version of the dataset one uses. The specific distribution of normal and attack connections in the NSL-KDD dataset can also vary, with some versions having a higher proportion of attack records than others (see Table 2).

### 6.2. Data Preprocessing

The realistic data contains anomalous and redundant instances due to the heterogeneity of the platforms, which may have a negative impact on classification accuracy. As a result, data preprocessing is the most time-consuming and crucial step in the data mining process [43]. In our work, as part of the preprocessing phase, we conducted data filtration, data transformation, and data normalization.

To create flow data, we started by combining the packets that match the values of a row in a CSV file with the row’s label values. Removing the dataset’s constant and redundant columns gives no meaningful classification information. In the CIC-IDS2017 dataset, for example, the ‘Fwd Header Length’ feature appears twice, while ‘Flow Packet/s’ has unusual values, like ‘Infinity’ and ‘NaN’. After that, we perform data filtration by inspecting the data type of each variable in the dataset and replacing any NA or empty data with zeroes.

Furthermore, various scales among features might affect classification performance. For instance, features with large numeric values, such as ‘Flow Duration’, could dominate the classifier’s model compared to features with lower numeric values, such as ‘Total Fwd Packets’. Normalization is, therefore, a scaling-down transformation that translates features to a normalized range. In our experiments, we applied the minimum–maximum method [44], which is defined as follows:(6)$$x_{\mathrm{scaled}\;}=\frac{x-\min(x)}{\max(x)-\min(x)}$$
where *x* is the original value, min is the minimum value of the column, and max is the maximum value of the column.

For preprocessing the data in NSL-KDD, which has multiple types and ranges of features, the process involves converting numeric features and normalizing the values. The three non-numeric features (protocol type, service, and flag) are transformed into numerical representations, known as one-hot vectors. For example, the “protocol type” feature, which has three categories (TCP, UDP, and ICMP), will be transformed into a one-hot vector. The normalization technique used is the min–max normalization method, which scales all numeric features to a range of [0, 1] to remove the impact of the feature value ranges in the input data.

### 6.3. Experimental Setup

The experiments were conducted on a desktop computer equipped with an Intel(R) core I9-7900X CPU running at 3.30 GHz, with 64 GB of RAM, using the Linux Ubuntu 16.04 operating system. The simulations were performed using PyTorch and scikit-learn, which are commonly used machine learning frameworks. Python was chosen as the programming language and the model was trained for 10,000 epochs with a batch size of 30. We used the Adam optimizer with a 0.0001 learning rate for both the generator and discriminator. The CICIDS2017 dataset used in the experiment contained 15 classes, but in this study, certain classes, such as brute force, structured query language (SQL) injection, and XSS attacks were grouped together and labeled as web attack classes, resulting in a total of 13 classes. For feature reduction using an autoencoder, 77 features were used as the input, and the optimal parameters were determined by adjusting the number of hidden layers.

### 6.4. Implementation of SGAN-IDS

The generator takes a noise vector as input and converts it into a fake training sample, which is subsequently passed via the discriminator. The discriminator uses both real samples (from the training data) and fake samples (produced by the generator) to distinguish between the two. The generator’s idea is to produce malicious traffic that can mislead the IDS. We preserve the functional features of an original attack to confirm that the generative algorithm’s output is actually an attack. The dimensions of hidden layers are set based on the best outcomes. The architecture of the model is detailed in Section 4.

The training input contains normal traffic records and malicious traffic records. The malicious records are fed into the generator and transformed into adversarial records after adding noise. The output of the first hidden layer generator will be forwarded to the attention layer, which acts as *X* number of units, and two heads will be defined in the model. The adversarial malicious records and normal records are predicted by the BlackBox IDS, using 79 features that have been preprocessed, and accurately predicting whether these data are benign or malicious traffic. For class identification, we adopt a threshold of 0.5 in this study. If σ≥0.5, IDS returns a value of 1 (attack); otherwise, if σ<0.5, the label is 0 (benign). After that, the discriminator uses the predicted labels to learn the BlackBox IDS. The generator and discriminator losses are estimated using the discriminator’s outputs and the IDS-predicted labels. Algorithm 1 describes the procedure for training the GAN.

### 6.5. Evaluation Metrics

The study’s findings are assessed using two criteria: detection rate (DR) and F1 score (F1) [45]. The attack detection rate is the ratio of the number of attack flows identified by IDS to the actual test attack data, as described in Equation (Equation 7). We examine the DR using both original attack data from the original dataset and adversarial attack data created by the SGAN-IDS generator. The F1 score is the harmonic mean of the precision score and recall, as shown in Equation (Equation 10). IDS’s ability to cover false positive and negative scenarios is measured using the F1 score. This statistic is used to assess the effectiveness of utilizing adversarial data in retraining.
(7)$$DetectionRate=\frac{\mathrm{Detected}\mathrm{Attack}}{\mathrm{all}\mathrm{Attack}}*100$$
(8)$$Precision=\frac{\mathrm{True}\mathrm{Positive}}{\mathrm{Total}\mathrm{Predicted}\mathrm{Positive}}$$
(9)$$Recall=\frac{\mathrm{True}\mathrm{Positive}}{\mathrm{Total}\mathrm{Actual}\mathrm{Positive}}$$
(10)$$F1=2\times \frac{\mathrm{Precision}*\mathrm{Recall}}{\mathrm{Precision}+\mathrm{Recall}}$$

### 6.6. Evaluation Results

Multiple machine learning algorithms have been used in IDSs based on relevant intrusion detection research [46]. We utilized five ML-based BlackBox IDS models to test the capability and generalization of our proposed model versus IDS. The algorithms include support vector machine (SVM), naïve Bayes (NB), multilayer perceptron (MLP), logistic regression (LR), and K-nearest neighbor (KNN). These models are often used as baseline approaches to verify enhanced intrusion detection systems.

We assess the detection performance of our proposed SGAN-IDS model by comparing it to state-of-the-art malicious traffic detectors. Before producing adversarial samples, the BlackBox IDS models are pre-trained with the training set.

In terms of determining any negative impacts of adversarial training, the experiment first evaluates the BlackBox IDS models on unchanged data, as shown under original traffic in Table 3. This table summarizes the results of the detection rates of the trained models in the unchanged test datasets both before (original traffic) and after (adversarial traffic), applying SGAN-IDS.

Without using SGAN-IDS, the models were able to identify DDoS flows with an average DR score of 97.87%. For the web attack, the average DR of the IDS-based models without SGAN-IDS was 97.1%. Similarly, the average DR of the models before applying SGAN-IDS was 97.40% for infiltration attacks. In total, the average DR of all models before using SGAN-IDS for all three types of attacks was 97.46%. In comparison, after including our system SGAN-IDS, the average DR percentages of the models dropped to 77.29%, 78.98%, and 89.53% for DDoS, web, and infiltration attacks, respectively. In summary, for all three attacks after using SGAN-IDS, the average DR is 81.93%, which is a decrease of 15.93% in comparison with the average DR percentages of the models without using SGAN-IDS (i.e., 97.46%)

The other criterion that we used to assess SGAN-IDS is the F1 score. Table 3 illustrates the F1 score for each attack type and ML model before (original traffic) and after (adversarial traffic) the inclusion of SGAN-IDS. Specifically, the average F1 score percentages of all the models without using SGAN-IDS were 97.19%, 96.77%, and 97.03% for DDoS, web, and infiltration attacks, respectively. On the other hand, after using SGAN-IDS to generate adversarial traffic flows, the average F1 score percentages of all the models decreased to 75.98% for DDoS, 76.82% for Web attack, and 88.50% for infiltration attack.

In summary, our system’s SGAN-IDS successfully generated adversarial traffic flows that resulted in decreasing the detection rate, precision, recall, and F1 score of five BlackBox ML-based IDS models.

Table 4 compares the performances of five different machine learning models (SVM, KNN, NB, LR, and LSTM) in classifying network traffic into three categories: DDoS, U2R, and probe. The evaluation metrics used are accuracy, precision, recall, and F1. The table contains two sections, the first section shows the performances of the models on original network traffic while the second section shows the performances on adversarial network traffic. For each model, the accuracy, precision, recall, and F1 are reported for the three categories of network traffic (DDoS, U2R, and probe). The values are expressed as percentages. For example, the accuracy of the SVM model on the original network traffic for the DDoS category is 97.87, its precision on the same category is 98.41, its recall is 95.36, and its F1 is 98.98.

It can be seen from the table that the performances of the models vary, depending on the type of network traffic, original or adversarial, and the category being considered. Regarding original network traffic, most models have high accuracy and F1 scores, but the performances drop significantly on adversarial network traffic.

### 6.7. Comparisons with State-of-the-Art Adversarial Techniques

In this comparison, various adversarial learning methods, besides GAN, were used to help traffic records avoid detection by an intrusion detection system (IDS). The proposed approach was compared with competitive baseline attack models, such as a JSMA attack, FGSM attack, DeepFool attack, and CW attack. The experiments used multilayer perceptrons as the intrusion detection systems, and the models and hyperparameters were kept consistent with previous work. The proposed approach outperformed all the baseline models by a wide margin in both malicious traffic categories. The results also showed differences between the various adversarial attack models, with the JSMA attack and CW attack being less effective. The proposed approach was also compared with a GAN-based method, and it was found to be more effective at evading the IDS while preserving traffic functionality (see Table 5).

## 7. Conclusions

In this paper, we address the challenge of crafting adversarial network flows that can evade detection via BlackBox IDSs. To confront this issue, we introduce the SGAN-IDS framework, which leverages the self-attention mechanism and GANs to enhance the resilience of ML models against attack detection. Our evaluation of SGAN-IDS utilizes the CICIDS2017 dataset, which encompasses a diverse range of attack types. Experimental outcomes reveal that SGAN-IDS diminishes the average detection rate of five ML-based IDSs from 97.46% to 81.93%, marking a reduction of 15.93%. The aforementioned results underscore the robustness and broad applicability of our proposed methodology.

## Figures and Tables

**Figure 1 sensors-23-07796-f001:**
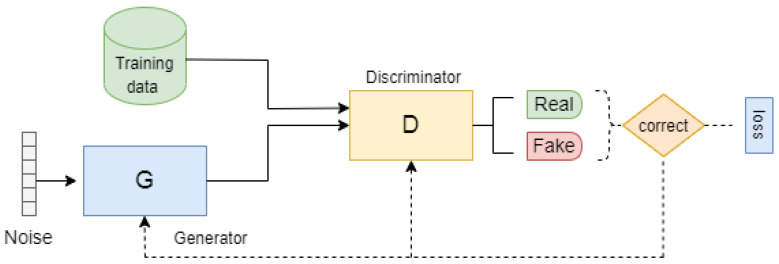
Architecture of the generative adversarial network (GAN).

**Figure 3 sensors-23-07796-f003:**
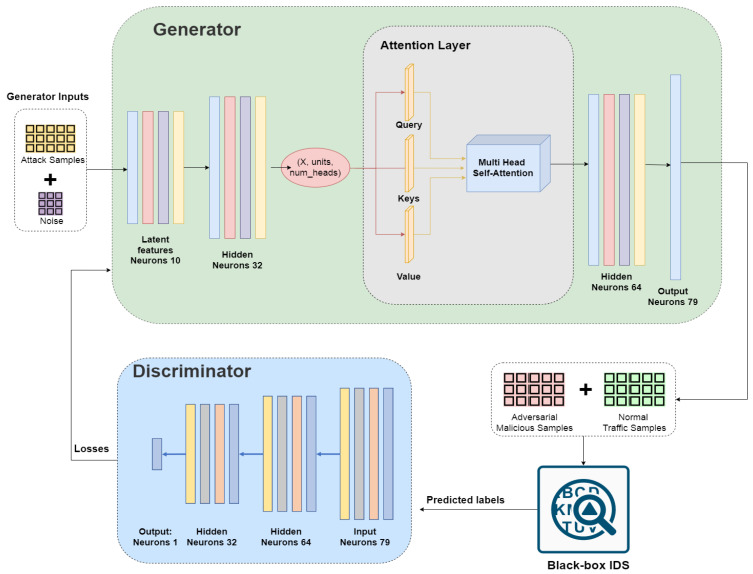
The architecture of the SGAN-IDS.

**Figure 4 sensors-23-07796-f004:**
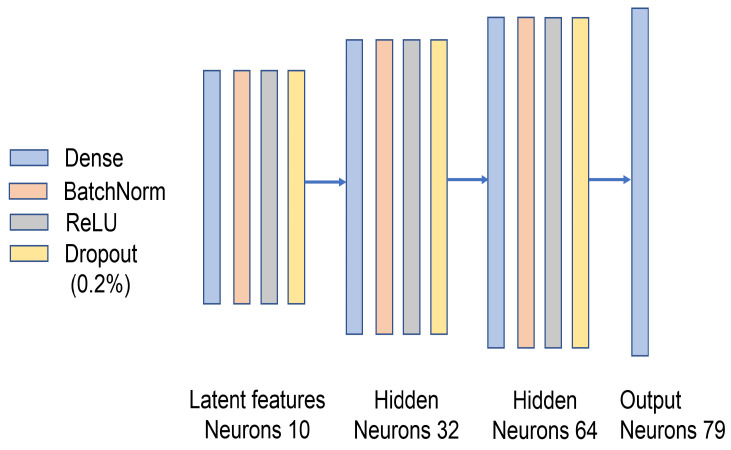
Design of the generator.

**Figure 5 sensors-23-07796-f005:**
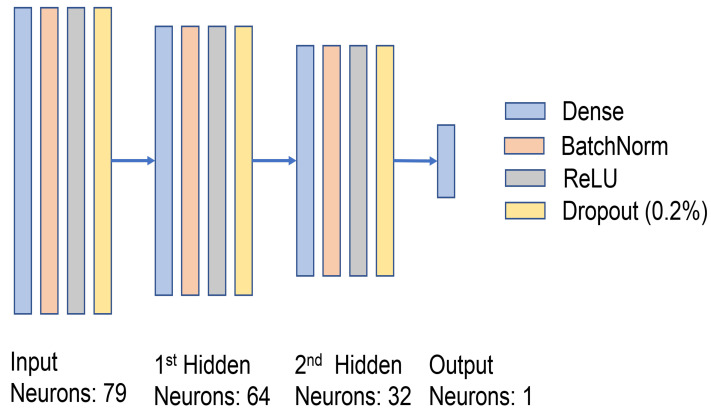
Design of the discriminator.

**Figure 6 sensors-23-07796-f006:**
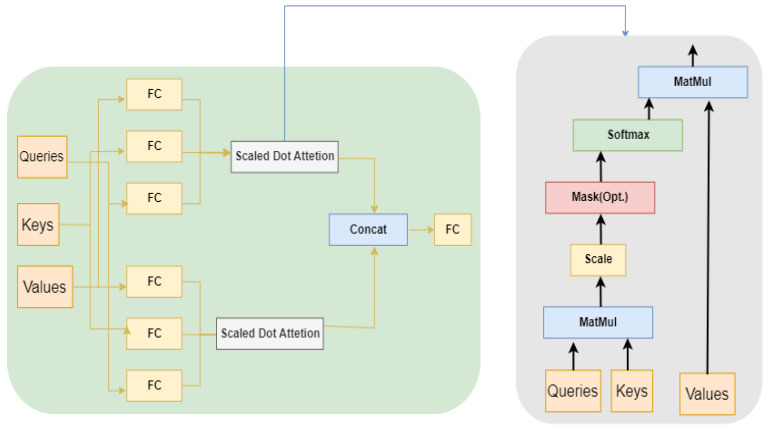
Design of the multi-head self-attention model.

**Table 1 sensors-23-07796-t001:** CICIDS2017 dataset statistics.

Class		Flow Count	Percentage	Training 70%	Testing 30%
**BENIGN**	BENIGN	2,273,097	76.75%	1,591,167	681,929
**DoS**	DDoS	231,073	7.802%	161,751	69,321
	Heartbleed	11	0.0003%	7	3
	DoS slowloris	5796	0.1957%	4057	1738
	DoS GoldenEye	10,293	0.3475%	3087	7205
	DoS SlowHTTPTest	5499	0.1856%	3849	1649
	DoS Hulk	231,073	0.0392%	161,751	69,321
**Web Attack**	SQL Injection	5796	0.2121%	4057	1738
	Brute Force	7938	0.2906%	5556	2381
	XSS	5897	0.2158%	4127	1769
**Infiltration**	Infiltration	10,293	0.3768%	7205	3087
**Port Scan**	Port scan	158,930	5.8184%	111,251	47,679
**Brute Force**	FTP-Patator	1769	0.29061%	1238	530
	SSH-Patator	5897	0.2158%	4127	1769
**Bot**	Bot	1966	0.0719%	1376	589

**Table 2 sensors-23-07796-t002:** NSL-KDD dataset statistics.

Class	Flow Count	Training 70%	Testing 30%
**Normal**	77,054	53,937	23,116
**DoSS**	53,387	37,370	16,016
**Probe**	14,077	9853	4223
**R2L**	3880	2716	1164
**U2R**	119	83	35

**Table 3 sensors-23-07796-t003:** Detection rate of the SGAN-IDS model against the state-of-the-art machine learning-based IDSs on the CICIDS-2017 dataset.

Original Traffic												
**Model**	**Accuracy**			**Precision**			**Recall**			**F1**		
	**DDoS**	**Web Attack**	**Infiltration**	**DDoS**	**Web Attack**	**Infiltration**	**DDoS**	**Web Attack**	**Infiltration**	**DDoS**	**Web Attack**	**Infiltration**
**SVM**	98.31	97.11	99.33	97.11	96.98	98.00	97.26	97.50	97.33	97.18	97.70	97.63
**KNN**	97.34	94.22	93.33	96.34	93.29	94.67	95.98	94.99	94.89	96.15	94.65	94.79
**NB**	98.00	97.88	98.44	97.60	97.54	97.74	97.12	97.20	97.14	97.35	97.30	97.54
**LR**	97.34	97.40	96.63	97.30	96.33	95.53	97.98	97.11	96.83	97.63	96.90	96.45
**LSTM**	98.34	98.88	99.23	98.11	97.78	97.93	97.98	97.88	98.93	97.65	97.80	98.73
**Adversarial Traffic**												
**Model**	**Accuracy**			**Precision**			**Recall**			**F1**		
	**DDoS**	**Web Attack**	**Infiltration**	**DDoS**	**Web Attack**	**Infiltration**	**DDoS**	**Web Attack**	**Infiltration**	**DDoS**	**Web Attack**	**Infiltration**
**SVM**	73.58	83.52	89.44	73.00	81.00	88.94	71.38	81.00	89.44	72.18	81.00	88.74
**KNN**	64.55	64.55	84.35	62.76	61.59	83.15	64.11	61.00	82.00	63.65	61.35	82.11
**NB**	89.43	84.44	95.43	88.63	82.94	93.83	87.73	82.44	94.93	87.93	82.64	94.83
**LR**	84.22	74.72	82.77	82.22	73.62	81.37	82.22	72.52	81.77	82.75	72.98	81.47
**LSTM**	74.66	87.66	95.64	72.96	85.65	94.77	73.86	86.66	95.90	73.40	86.11	95.33

**Table 4 sensors-23-07796-t004:** Detection rate of the SGAN-IDS model against the state-of-the-art machine learning-based IDSs on the NSL-KDD dataset.

Original Traffic												
**Model**	**Accuracy**			**Precision**			**Recall**			**F1**		
	**DDoS**	**U2R**	**Probe**	**DDoS**	**U2R**	**Probe**	**DDoS**	**U2R**	**Probe**	**DDoS**	**U2R**	**Probe**
**SVM**	97.87	97.99	97.93	98.41	96.98	98.00	95.36	92.59	96.93	98.98	95.00	96.63
**KNN**	95.24	97.22	96.33	96.74	93.29	94.67	95.98	94.99	94.89	96.15	94.65	94.79
**NB**	98.80	97.00	94.44	97.60	98.54	94.74	95.02	97.20	95.94	96.75	98.90	95.50
**LR**	96.14	96.98	98.63	97.30	96.33	98.83	97.98	97.11	96.83	97.63	96.90	96.45
**LSTM**	98.50	97.48	97.93	96.16	96.78	97.93	96.98	92.08	95.95	95.95	92.90	97.73
**Adversarial Traffic**												
**Model**	**Accuracy**			**Precision**			**Recall**			**F1**		
	**DDoS**	**U2R**	**Probe**	**DDoS**	**U2R**	**Probe**	**DDoS**	**U2R**	**Probe**	**DDoS**	**U2R**	**Probe**
**SVM**	51.59	56.11	49.44	53.62	41.00	58.94	53.65	31.00	49.44	42.18	33.65	53.73
**KNN**	44.55	44.93	34.35	22.76	21.59	36.11	44.11	41.00	32.00	43.65	31.35	52.11
**NB**	42.64	34.44	35.43	82.44	46.11	31.90	37.73	52.44	54.93	37.93	32.64	44.53
**LR**	24.22	45.64	22.77	22.22	23.62	31.37	22.22	32.52	41.77	22.75	42.98	31.37
**LSTM**	22.44	31.77	33.40	73.62	35.65	31.47	43.86	23.65	35.90	33.65	21.47	35.34

**Table 5 sensors-23-07796-t005:** Comparisons with state-of-the-art adversarial techniques.

Attack	Baseline	FGSM	JSMA	DeepFool	SGAN-IDS
**DoSS**	98.50	36.33	67.33	26.64	22.44
**Probe**	97.48	43.23	51.77	30.23	31.77
**U2R**	97.93	45.98	41.47	37.98	33.40

## Data Availability

Not applicable.

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
