# Peer review of "SGAN-IDS: Self-Attention-Based Generative Adversarial Network against Intrusion Detection Systems"

_sensors, 2023, doi:10.3390/s23187796_

Round 1

Reviewer 1 Report

The article is devoted to the issue of Intrusion Detection. The topic of the article is relevant. The structure of the article does not correspond to that adopted in the MDPI for research articles (Introduction (including analysis of analogues), Models and methods, Results, Discussion, Conclusions). The level of English is acceptable. The article is easy to read. The figures in the article are of acceptable quality. The article cites 47 sources, some of which are not relevant. The References section is sloppy.

The following comments and recommendations can be formulated on the material of the article:

1. Intrusion Detection and Prevention System (IDPS) are intrusion detection and prevention systems. In essence, IDPS monitors transit and local traffic for scanning and attack attempts, correlating them with available signatures. If the traffic is "malicious" - it is blocked. It is IDPS, and not IPS, that is the focus of attention. I ask the authors to reflect this moment in the text of the flock.

1. IDPS can be divided into two classes - NIDS (Network Intrusion Detection System) and HIDS (Host-based Intrusion Detection). The former monitor network traffic, while the latter analyze host events, including incoming and outgoing traffic within systems. As the name implies, NIDS should be installed on hosts that control traffic, while HIDS are more suitable for endpoint hosts with local services. You can read more about the classification and features of IDPS systems at the Selectel Academy. I will mention two popular representatives of IDPS systems: - Suricata (as an instance in Ubuntu 20.04) is a high-performance traffic analysis and threat detection software; - Snort (as a package in pfSense 2.6.0) is one of the most popular open source IDPS. I ask the authors to spawn their offer with these analogues.

3. One of the methods for identifying compromised hosts is network traffic analysis. There are four main methods of analysis: - Shallow Packet Inspection (hereinafter referred to as SPI) – the technology works at the link and network levels of the OSI model (The Open Systems Interconnection model) and checks only the packet header; - Medium Packet Inspection (hereinafter referred to as MPI) - the technology analyzes sessions and communication sessions that were initialized by applications; - Lightweight Payload Inspection (hereinafter referred to as LPI) - the technology analyzes the first bytes of packet headers, as well as sessions and communication sessions; - Deep Packet Inspection (hereinafter referred to as DPI) - this technology analyzes the belonging of a packet stream to a specific application, and is often used to manage and effectively distribute network load. In the article, I will focus on this type of solution. How do the authors position their solution within this classification?

4. Solutions from manufacturers CheckPoint, Palo Alto and FortiNet provide a comprehensive approach to protecting computer networks of any complexity and volume of analyzed traffic. The functionality is varied and extensive, both in terms of network security and in terms of network functionality. CheckPoint CheckUP and the Cyber Threat Assessment Program (FortiNet) provide well-researched reports, but have a limitation on the duration of use, since the goals of these solutions are to demonstrate the security of the infrastructure once. Security Onion can be used as a permanent protection tool with almost no additional financial costs, but since the product is open source, there will be more labor costs for the specialist (s) implementing or maintaining the solution. These are serious competitors with their own audience of users. And what are the prospects for the implementation of their research have the authors?

-

Author Response

First, we would like to thank the editor and the reviewers for a careful reading of the manuscript and the positive assessment of our work. Below is a point-by-point response to the comments of the reviewer

  1. Response to Reviewers’ Comments

Comments from Reviewer 1

Thank you for taking the time to leave your feedback. We appreciate your time and effort in advocating for us. The following comments and recommendations can be formulated on the material of the article:

  • Comment: The article is devoted to the issue of Intrusion Detection. The topic of the article is relevant. The structure of the article does not correspond to that adopted in the MDPI for research articles (Introduction (including analysis of analogues), Models and methods, Results, Discussion, Conclusions). The level of English is acceptable. The article is easy to read. The figures in the article are of acceptable quality. The article cites 47 sources, some of which are not relevant. The References section is sloppy.

Response: Thank you for taking the time to review our article and we appreciate your feedback and insights on the structure and content of the article. We acknowledge your observation regarding the structure of the article not aligning with the format adopted by MDPI for research articles. We will make the necessary revisions to ensure that the article adheres to the recommended structure, including the sections: Introduction, Models and Methods, Results, Discussion, and Conclusions.

  • Comment: Intrusion Detection and Prevention System (IDPS) are intrusion detection and prevention systems. In essence, IDPS monitors transit and local traffic for scanning and attack attempts, correlating them with available signatures. If the traffic is "malicious" - it is blocked. It is IDPS, and not IPS, that is the focus of attention. I ask the authors to reflect this moment in the text of the flock...

Response: Thank you for pointing out the distinction between Intrusion Detection and Prevention System (IDPS) and Intrusion Prevention System (IPS). We understand the significance of emphasizing IDPS, given its comprehensive approach to monitoring both transit and local traffic, correlating potential threats with existing signatures, and taking necessary actions when malicious activity is detected. We will ensure that the text is revised to accurately reflect the focus on IDPS and its functionalities. We updated the manuscript to use "IDS" specifically when referring to the systems being tested against adversarial attacks. This helps clarify that the intrusion detection systems are being discussed rather than prevention systems. Thank you again for catching this nuance - it will help avoid confusion and make the focus on adversarial evasion of IDS clear to readers.

  • Comment: IDPS can be divided into two classes - NIDS (Network Intrusion Detection System) and HIDS (Host-based Intrusion Detection). The former monitor network traffic, while the latter analyze host events, including incoming and outgoing traffic within systems. As the name implies, NIDS should be installed on hosts that control traffic, while HIDS are more suitable for endpoint hosts with local services. You can read more about the classification and features of IDPS systems at the Selectel Academy. I will mention two popular representatives of IDPS systems: - Suricata (as an instance in Ubuntu 20.04) is a high-performance traffic analysis and threat detection software; - Snort (as a package in pfSense 2.6.0) is one of the most popular open source IDPS. I ask the authors to spawn their offer with these analogues.

Response: Thank you for your insightful comment and for highlighting the distinction between NIDS and HIDS. We appreciate the references to Suricata and Snort as popular representatives of IDPS systems. To address your comment: Our model is primarily focused on the NIDS (Network Intrusion Detection System) category which in our case  base on machine learning approach. It is designed to monitor network traffic for signs of malicious activity and potential threats, similar to how Suricata operates. While we acknowledge the importance and utility of HIDS in certain scenarios, our solution is tailored towards providing comprehensive network-level surveillance and threat detection.

Comparatively, our model improve IDSs’ resilience which generates malicious feature records for adversarial attacks on intrusion detection systems. The goal of is to deceive the defense systems and carry out successful evasion in real-world networks.

  • Comment: One of the methods for identifying compromised hosts is network traffic analysis. There are four main methods of analysis: - Shallow Packet Inspection (hereinafter referred to as SPI) – the technology works at the link and network levels of the OSI model (The Open Systems Interconnection model) and checks only the packet header; - Medium Packet Inspection (hereinafter referred to as MPI) - the technology analyzes sessions and communication sessions that were initialized by applications; - Lightweight Payload Inspection (hereinafter referred to as LPI) - the technology analyzes the first bytes of packet headers, as well as sessions and communication sessions; - Deep Packet Inspection (hereinafter referred to as DPI) - this technology analyzes the belonging of a packet stream to a specific application, and is often used to manage and effectively distribute network load. In the article, I will focus on this type of solution. How do the authors position their solution within this classification?

  → Response:  Thank you for your detailed comment on the various methods of network traffic analysis, ranging from Shallow Packet Inspection (SPI) to Deep Packet Inspection (DPI). We appreciate the depth and clarity you provided on this classification.

In response to your query on how our solution fits within this classification: Our proposed SGAN-IDS framework operates at a different layer of abstraction compared to the packet inspection methods you've outlined. Instead of focusing on the packet content or its specific attributes, our model primarily targets the feature space of machine learning-based Intrusion Detection Systems (IDS). The essence of SGAN-IDS is to generate adversarial traffic records that can evade detection by black-box ML-based IDS models. This is achieved by leveraging the Generative Adversarial Network (GAN) paradigm, where the generator creates adversarial malicious traffic records, and the discriminator learns from the outputs of the black-box IDS. The goal is not to analyze or inspect the packets per se, but to understand and exploit the decision boundaries of ML-based IDS models, thereby deceiving them.

While the methods you mentioned (SPI, MPI, LPI, DPI) are crucial for the initial stages of traffic analysis and filtering, our framework comes into play at a subsequent stage, where the IDS has already processed the traffic and is about to make a decision. SGAN-IDS aims to challenge and test the robustness of these IDS models by generating traffic that they fail to detect as malicious. In essence, while the methods you've described are about "reading" the traffic, our SGAN-IDS is about "writing" or "crafting" traffic in such a way that it evades detection. We hope this clarifies the positioning of our solution in relation to the classification you've provided.

  • Comment: Solutions from manufacturers CheckPoint, Palo Alto and FortiNet provide a comprehensive approach to protecting computer networks of any complexity and volume of analyzed traffic. The functionality is varied and extensive, both in terms of network security and in terms of network functionality. CheckPoint CheckUP and the Cyber Threat Assessment Program (FortiNet) provide well-researched reports, but have a limitation on the duration of use, since the goals of these solutions are to demonstrate the security of the infrastructure once. Security Onion can be used as a permanent protection tool with almost no additional financial costs, but since the product is open source, there will be more labor costs for the specialist (s) implementing or maintaining the solution. These are serious competitors with their own audience of users. And what are the prospects for the implementation of their research have the authors?

Response: Thank you for pointing out the capabilities and nuances of established network security solutions from manufacturers like CheckPoint, Palo Alto, and FortiNet. Indeed, these solutions offer a comprehensive approach to network security, each with its own strengths, limitations, and target audience. As you rightly mentioned, while CheckPoint CheckUP and FortiNet's Cyber Threat Assessment Program provide detailed reports, they are primarily designed for one-time demonstrations of infrastructure security. On the other hand, Security Onion, being open-source, offers a cost-effective solution but demands more hands-on expertise for implementation and maintenance.

Regarding  the Generative Adversarial Network (GAN) framework for Intrusion Detection Systems (IDS), the prospects are multifaceted:

  • Innovation in IDS: The proposed SGAN-IDS framework introduces a novel approach to enhancing the robustness of IDSs against adversarial attacks. By generating synthetic intrusion flows, it aims to simulate potential future attacks, thereby improving the accuracy of IDSs.

  • Filling the Gap: While established solutions focus on comprehensive network security, our model specifically targets the challenge of adversarial synthetic network flows, which can decrease IDS detection rates. This niche focus can complement existing solutions by bolstering their defenses against such advanced threats.

  • Cost-Effectiveness: Similar to open-source solutions like Security Onion, the GAN-based approach might offer a cost-effective alternative or addition to commercial solutions, especially for organizations with the expertise to implement and maintain it.

  • Integration Potential: The research could be integrated into existing security solutions to enhance their capabilities against adversarial attacks. This would broaden the applicability of the research and make it relevant to a wider audience.

  • Continuous Evolution: Given the dynamic nature of cybersecurity threats, the GAN-based approach offers a framework that can continuously evolve and adapt to new types of attacks, ensuring long-term relevance.

Reviewer 2 Report

This study presented a framework called SSGAN-IDS that uses adversarial training to make ML-based IDSs more resilient to attack detection. The reviewer has the following comments:

1)  1. The authors need to highlight theory contributions of this study.

2)  The motivation on why to propose such a framework and strategy in real-world applications should be clearly emphasized.

3) Definitions should be given for some symbols that are used first time in this paper, for example, in Eq. (2).

4) Update the recent reference related to this work: Event-triggered H∞ filtering for T-S fuzzy-model-based nonlinear networked systems with multisensors against dos attacks

5) Several syntax errors necessitate thorough proofreading, and adjustments are needed for both the image and font sizes across numerous figures.

6) How can you get the statistic data from Figure 7 in page 12.

In abstract, Correction: Evaluation results show that SGAN-IDS has successfully constructed adversarial flows of different attack types and has decreased the detection rate of all five IDSs by an average of 15.93%, which indicates the robustness and wide feasibility of the proposed model.

 In Conclusions, Correction: To solve this problem, we have built the SGAN-IDS framework that utilizes a self-attention mechanism and GANs to make ML models more resilient to attack detection.

Author Response

Comments from Reviewer 2

Thank you for taking the time to leave your feedback. We appreciate your time and effort in advocating for us.

  • Comment: The authors need to highlight theory contributions of this study..

Response: We apologize for not being more clear. we propose a new Generative Adversarial Network (GAN) framework, named SGAN-IDS, that generates malicious feature records for adversarial attacks on intrusion detection systems. The goal of the framework is to deceive the defense systems and carry out successful evasion in real-world networks. The framework, which builds upon the GANs and self-attention mechanisms, includes a generator that creates adversarial malicious traffic records and a discriminator that learns from the black-box intrusion detection system (IDS) by analyzing its real-time outputs and providing feedback for the generator’s training. The IDS outputs can be obtained by querying it with traffic records. Experimental results have shown that SGAN-IDS successfully generated adversarial traffic flows resulting in decreasing the detection rate, precision, recall, and F1 score of 5 black-box ML-based IDS models.

  • Comment: The motivation on why to propose such a framework and strategy in real-world applications should be clearly emphasized.

Response: Thank you for your feedback. We recognize the importance of clearly articulating the motivation behind our proposed framework, especially in the context of real-world applications. Here's a concise rationale:

  • Evolving Threat Landscape: With cyber threats becoming more sophisticated, traditional IDSs often fall short. Our framework addresses the need for advanced solutions tailored to contemporary challenges.

  • Synthetic Data Vulnerabilities: Current IDSs can be bypassed by adversarial synthetic network flows. Our proposal directly confronts this vulnerability, aiming to enhance detection capabilities.

  • Harnessing Deep Learning: The capabilities of GANs in understanding complex data distributions present an opportunity to revolutionize cybersecurity. Our framework leverages this potential for improved intrusion detection.

  • Proactive Approach: Our strategy emphasizes not just reacting to threats but proactively simulating potential attacks, ensuring IDSs are better prepared for real-world threats.

  • AI in Cybersecurity: As AI becomes integral to security solutions, it's crucial to address its potential vulnerabilities. Our framework seeks to turn AI's challenges into strengths, enhancing overall security.

We hope this provides clarity on the motivation behind our SGAN-IDS framework and its relevance to real-world cybersecurity challenges. We appreciate your valuable insights and will ensure this motivation is emphasized in our revised manuscript.

  • Comment: Definitions should be given for some symbols that are used first time in this paper, for example, in Eq. (2).

Response:: Thank you for pointing that out. We have indeed provided definitions for all symbols when they are introduced in the paper, including those in Eq. (2). We'll ensure that these definitions are more prominently highlighted in the revised manuscript for better clarity.

  • Comment: Update the recent reference related to this work: Event-triggered H∞ filtering for T-S fuzzy-model-based nonlinear networked systems with multisensors against dos attacks

Response: Thank you for your feedback on our manuscript. We've updated it with the most recent references related to event-triggered H∞ filtering against DoS attacks in T-S fuzzy-model-based systems. We believe these additions enhance the paper's relevance and depth.

  • Comment: Several syntax errors necessitate thorough proofreading, and adjustments are needed for both the image and font sizes across numerous figures.

Response: Thank you for bringing this to our attention. We have conducted a comprehensive proofreading to address the syntax errors you mentioned. Additionally, adjustments have been made to both the image and font sizes in the figures to ensure consistency and clarity.

Comments on the Quality of English Language

In abstract, Correction: Evaluation results show that SGAN-IDS has successfully constructed adversarial flows of different attack types and has decreased the detection rate of all five IDSs by an average of 15.93%, which indicates the robustness and wide feasibility of the proposed model.

 In Conclusions, Correction: To solve this problem, we have built the SGAN-IDS framework that utilizes a self-attention mechanism and GANs to make ML models more resilient to attack detection.

Response: Thank you for bringing this to our attention. We have conducted a comprehensive proofreading to address the syntax errors you mentioned

Round 2

Reviewer 1 Report

I formulated the following remarks to the basic version of the article:

1. Intrusion Detection and Prevention System (IDPS) are intrusion detection and prevention systems. In essence, IDPS monitors transit and local traffic for scanning and attack attempts, correlating them with available signatures. If the traffic is "malicious" - it is blocked. It is IDPS, and not IPS, that is the focus of attention. I ask the authors to reflect this moment in the text of the flock.

2. IDPS can be divided into two classes - NIDS (Network Intrusion Detection System) and HIDS (Host-based Intrusion Detection). The former monitor network traffic, while the latter analyze host events, including incoming and outgoing traffic within systems. As the name implies, NIDS should be installed on hosts that control traffic, while HIDS are more suitable for endpoint hosts with local services. You can read more about the classification and features of IDPS systems at the Selectel Academy. I will mention two popular representatives of IDPS systems: - Suricata (as an instance in Ubuntu 20.04) is a high-performance traffic analysis and threat detection software; - Snort (as a package in pfSense 2.6.0) is one of the most popular open source IDPS. I ask the authors to spawn their offer with these analogues.

3. One of the methods for identifying compromised hosts is network traffic analysis. There are four main methods of analysis: - Shallow Packet Inspection (hereinafter referred to as SPI) – the technology works at the link and network levels of the OSI model (The Open Systems Interconnection model) and checks only the packet header; - Medium Packet Inspection (hereinafter referred to as MPI) - the technology analyzes sessions and communication sessions that were initialized by applications; - Lightweight Payload Inspection (hereinafter referred to as LPI) - the technology analyzes the first bytes of packet headers, as well as sessions and communication sessions; - Deep Packet Inspection (hereinafter referred to as DPI) - this technology analyzes the belonging of a packet stream to a specific application, and is often used to manage and effectively distribute network load. In the article, I will focus on this type of solution. How do the authors position their solution within this classification?

4. Solutions from manufacturers CheckPoint, Palo Alto and FortiNet provide a comprehensive approach to protecting computer networks of any complexity and volume of analyzed traffic. The functionality is varied and extensive, both in terms of network security and in terms of network functionality. CheckPoint CheckUP and the Cyber Threat Assessment Program (FortiNet) provide well-researched reports, but have a limitation on the duration of use, since the goals of these solutions are to demonstrate the security of the infrastructure once. Security Onion can be used as a permanent protection tool with almost no additional financial costs, but since the product is open source, there will be more labor costs for the specialist (s) implementing or maintaining the solution. These are serious competitors with their own audience of users. And what are the prospects for the implementation of their research have the authors?

The authors responded to all comments. I liked these answers. I support the publication of the current version of the article. I wish the authors creative success.